# Infrared Absorption Efficiency Enhancement of the CMOS Compatible Thermopile by the Special Subwavelength Hole Arrays

**Yun-Ying Yeh [1], Chih-Hsiung Shen [2] and Chi-Feng Chen [1],***

[1] Department of Mechanical Engineering, National Central University, Taoyuan City 32001, Taiwan; j9038592@ms38.hinet.net

[2] Department of Mechatronics Engineering, National Changhua University of Education, Changhua 50007, Taiwan; hilbert@cc.ncue.edu.tw

\* Correspondence: ccf@cc.ncu.edu.tw; Tel.: +886-3-426-7308

**Abstract:** The infrared absorption efficiency (IAE) enhancement of the complementary-metal-oxide-semiconductorCMOS compatible thermopile with special subwavelength hole arrays in an active area was numerically investigated by the finite-difference time-domain method. It was found that the absorption efficiency of that thermopile was enhanced when the subwavelength rectangular-hole array added extra rectangular-columnar or ellipse-columnar structures in the hole array. The simulation results show that the IAEs of the better cases for the three types of rectangular columns and three ellipse columns were increased by 14.4% and 15.2%, respectively. Such special subwavelength hole arrays can be improved by the IAE of the CMOS compatible thermopile.

**Keywords:** subwavelength; subwavelength hole arrays; thermopile; CMOS-MEMS; infrared radiation; infrared sensors

---

## 1. Introduction

With the advancement of computer-aided design and micro/nano-fabrication technologies [1], optical elements with functional micro-/nano-structures have been successfully used to improve the performance of components or modules, such as light-emitting diodes [2,3], photodetectors [4], solar systems [5,6], displays [7], and glass components [8]. The anti-reflective optical film having subwavelength structure arrays on its surface, and replicated by the use of a roll to roll micro-replication process is numerically and experimentally investigated [9,10]. The experimental results show that the Fresnel reflection on the interface surface is obviously suppressed when the subwavelength structure arrays exist on the surface of the optical film. In addition, metal films or doped silicon wafers with subwavelength hole arrays (SHAs) have been proposed to enhance the transmission [11–15]. Ebbesen et al. discovered the optical transmission of subwavelength cylindrical cavities in metal films could be significantly enhanced [11]. For such optically thick metal films, the zero-order transmission spectra are clearly related to the geometry of the hole array [11,12]. The transmission of terahertz radiation through highly doped silicon wafers with SHAs has been experimentally investigated [14,15]. It was found that the transmission is significantly enhanced, and the enhancement is related to the hole size and array thickness. We demonstrate extraordinary THz transmission of an array of subwavelength apertures patterned on ultrathin highly doped silicon by reactive ion etching. Additionally, several subwavelength hole arrays structured in the active area of the complementary-metal-oxide-semiconductorCMOS compatible thermopile are investigated [16]. It is numerically and experimentally shown that the measurement results are consistent with that of the simulation results, and the infrared absorption efficiency (IAE) is significantly enhanced. There is an interesting phenomenon to be discovered; the

rectangular column (RC) or ellipse column (EC) in a rectangular hole, which can be enhanced by the IAE of that thermopile for the best case in [16] (the rectangular type with the hole length 15 μm, the hole width 3.5 μm, and the array thickness 3.5 μm).

In this study, we investigated several special subwavelength columnar structures in the rectangular hole of the best case of [16], to enhance the infrared absorption efficiency of CMOS compatible thermopiles. Using the finite-difference time-domain (FDTD) method, we researched the best geometry of the extra subwavelength columnar structures (ESCS). It was obtained that for the three types of RCs and three ECs, the IAE enhancements of those thermopiles are 14.40% and 15.21%, respectively.

## 2. Simulation Method

The sketch of the proposed CMOS compatible thermopile is shown in Figure 1. Here $G_o$ is the total thermal conductance, $T_h$ is the hot junction temperature, and $T_a$ is the ambient temperature. The infrared radiation is absorbed in the active area on the front-side of a thermopile. The fabrication of the CMOS compatible thermopiles with SHA was considered as it uses the 0.35 μm 2P4M CMOS-micro-electro-mechanical systems (MEMS) process in the Taiwan Semiconductor Manufacturing Company (TSMC) [16].

In the above premise, we designed a thermopile with various SHAs by using the FDTD method. The FDTD method is an accurate and available technique to study thermopiles with SHA [13]. The sketch of the simulation model is shown in Figure 2, where $n_o$ is the air refractive index and $n_s$ is the SiO$_2$ refractive index. Here we considered $n_0 = 1$ and $n_s = 1.42$. For the FDTD method, an artificial boundary condition was required to suppress reflections at the analysis windows. The perfectly matched layer (PML) (ABC) is an absorbing boundary condition and is used to truncate the computational domain without reflection [17,18]. A perfect matched layer (PML) was applied to decrease the error caused by simulated region boundaries.

To explore the effect of the CMOS compatible thermopile with those ESCSs in the rectangular hole of the best case in [16], we considered six ESCSs and looked for the best geometry of the ESCSs by using the FDTD method. The six ESCSs included one RC, two RCs, three RCs, one EC, two ECs, and three ECs, and the top-view sketch is shown in Figure 3. Geometric parameters of the rectangular hole taken from [16] (the best SHA case) are a hole length (in the x-axis direction) of 15 μm, a hole width (in the x-axis direction) of 3.5 μm, and an array thickness of 3.5 μm. One can see that, based on the requirements of structure and heat conduction, we added some connection structures to connect those ESCSs to the main structure and set its value to 0.8 μm. The structures can be fabricated by the etching of layers and substrates beneath the floating structures. For the rectangular column, the geometric dimensions in the x-axis and y-axis directions are Wx and Wy, respectively. For the ellipse column, the geometric dimensions in the x-axis and y-axis directions were $D_x$ and $D_y$, respectively.

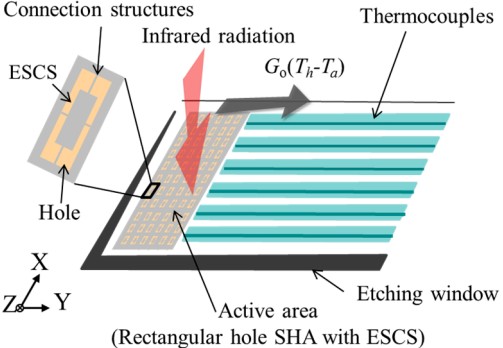

**Figure 1.** Sketch of the proposed CMOS compatible thermopile and heat conduction. SHA = subwavelength hole arrays, ESCS = extra subwavelength columnar structures.

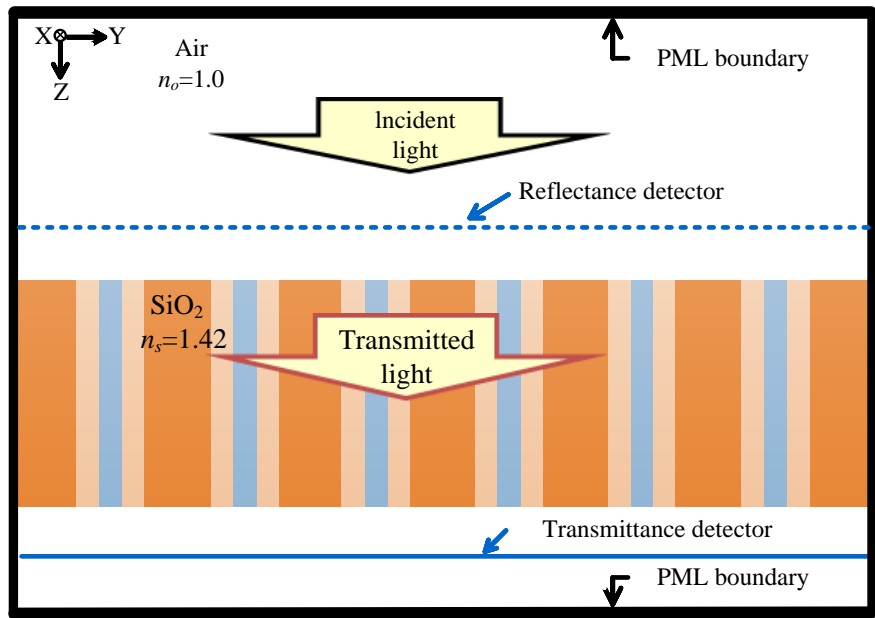

**Figure 2.** Sketch of a light-wave propagation through a CMOS compatible thermopile with SHA simulated by the finite-difference time-domain (FDTD) method.

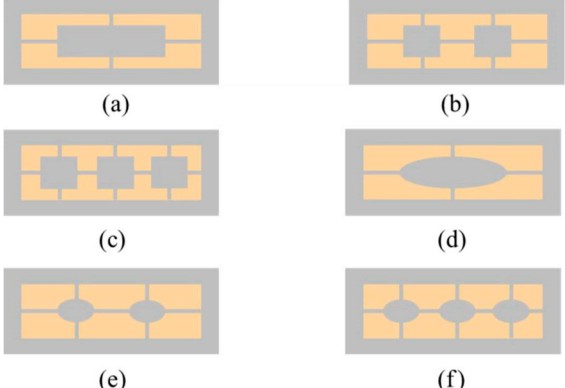

**Figure 3.** Top-view sketch of the six ESCSs, (**a**) one rectangular column (RC), (**b**) two RCs, (**c**) three RCs, (**d**) one ellipse column (EC), (**e**) two ECs, and (**f**) three ECs.

## 3. Results and Discussion

For convenience, the relative IAE is used to reveal the effect of the CMOS compatible thermopile with those ESCS and is defined as the IAE of that thermopile with the ESCS is relative to one without the ESCS and is written as:

$$Relative\ IAE = \frac{\text{the IAE of that thermopile with the ESCS}}{\text{the IAE of that thermopile without the ESCS}} \tag{1}$$

For the types of rectangular column, the variances of the relative IAEs with different $W_x$ and $W_y$ for the thermopiles with RC-type ESCSs at the target temperature of 75 °C are shown in Figure 4. It was seen that for the types of one RC, two RCs, and three RCs, the better relative IAEs were 1.128, 1.127, and 1.144 times, respectively. It was obtained that the $W_x$ and $W_y$ of the better case are 2.6 μm and 1.8 μm, respectively. For the types of ellipse column, the variances of the relative IAEs with different $W_x$ and $W_y$ for the thermopiles with EC-type ESCSs at the target temperature 75 °C are shown in Figure 5. One can see that for the types of one EC, two ECs, and three ECs, the better relative IAEs are 1.130, 1.132, and 1.152 times, respectively. It was obtained that the $D_x$ and $D_y$ of the better case were 3.2 μm and

1.5 μm, respectively. Those results of the better relative IAEs and the geometric parameters for the six ESCSs are listed in Table 1. Finally, the variances of the IAEs with the target temperature for the SHA thermopiles of (a) the best case in this study and (b) the best case in [16], and (c) the thermopile without SHA, are shown in Figure 6. One can see that the IAE of the CMOS compatible thermopiles was significantly enhanced when the subwavelength hole structure existed in the active area of the thermopiles, especially when the special structure in the rectangular hole array was added.

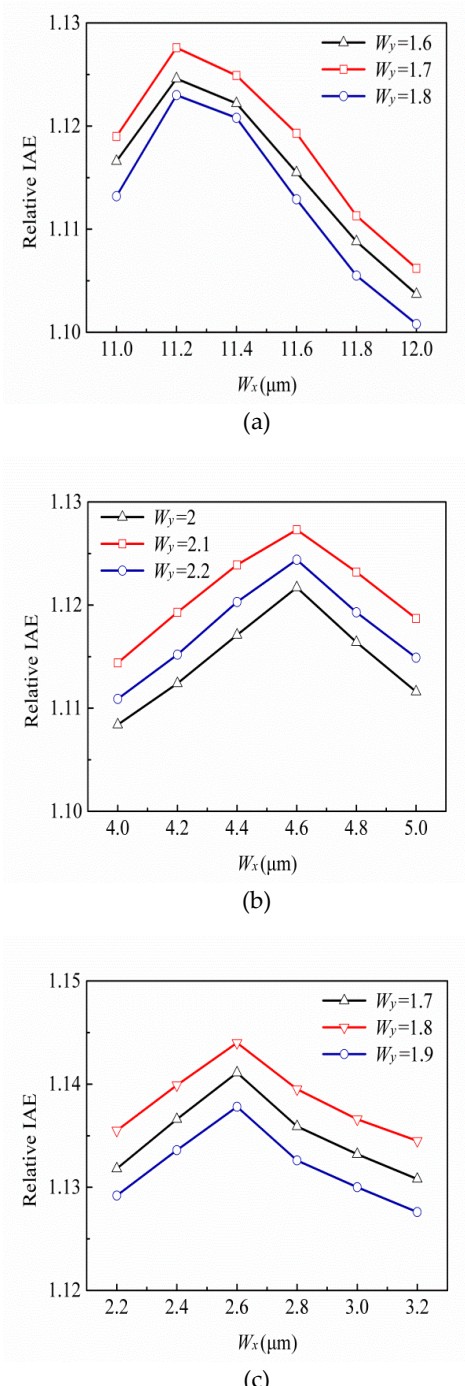

**Figure 4.** Variances of the relative infrared absorption efficiencies (IAEs) with different $W_x$ and $W_y$ for the thermopiles with RC-type ESCSs at the target temperature 75°, (**a**) one RC, (**b**) two RCs, and (**c**) three RCs.

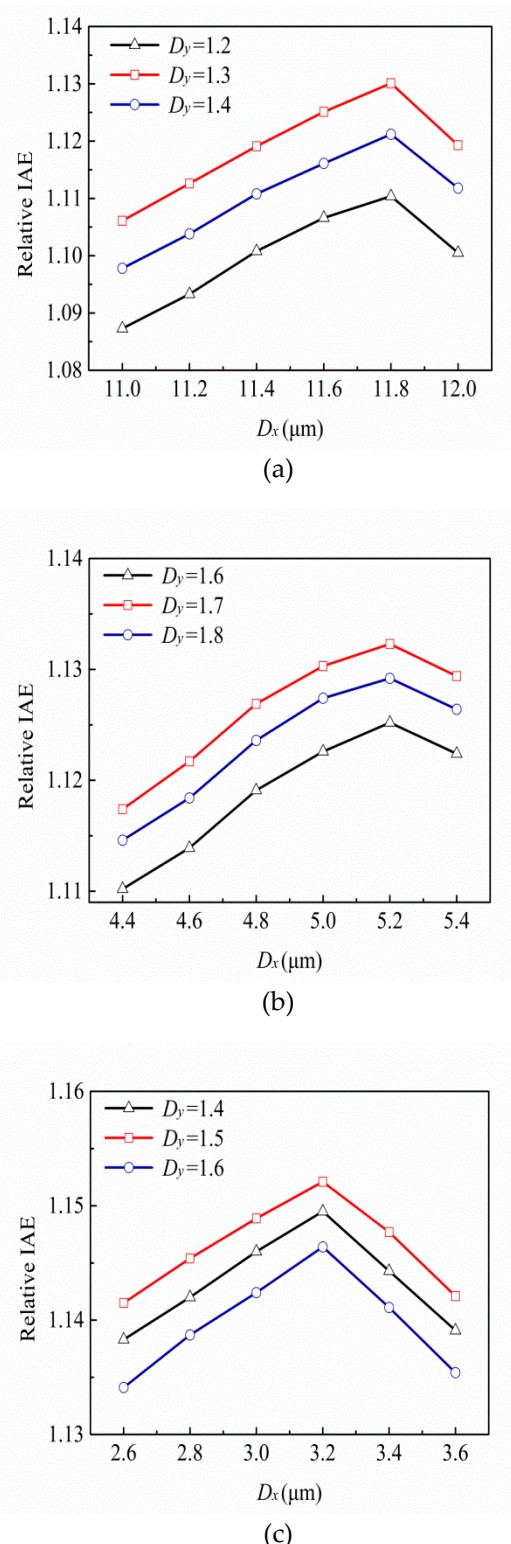

**Figure 5.** Variances of the relative IAEs with different $W_x$ and $W_y$ for the thermopiles with EC-type ESCSs at the target temperature 75 °C, (**a**) one EC, (**b**) two ECs, and (**c**) three ECs.

**Table 1.** Better relative IAEs and geometric parameters for the six ESCSs.

| ESCS Type | Geometric Parameters | | Relative IAE |
|---|---|---|---|
| | $W_x/D_x$ **(µm)** | $W_y/D_y$ **(µm)** | |
| One RC | 11.2 | 1.7 | 1.128 |
| Two RCs | 4.6 | 2.1 | 1.127 |
| Three RCs | 2.6 | 1.8 | 1.144 |
| One EC | 11.8 | 1.3 | 1.130 |
| Two ECs | 5.2 | 1.7 | 1.132 |
| Three ECs | 3.2 | 1.5 | 1.152 |

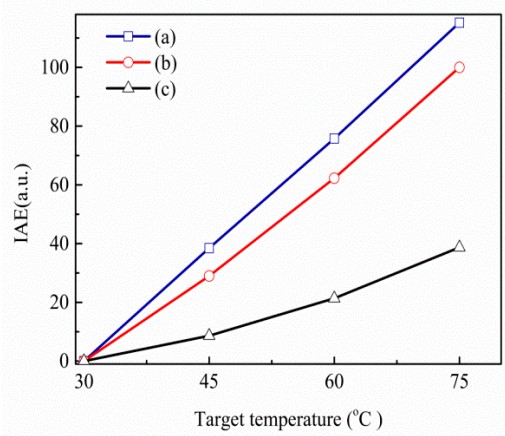

**Figure 6.** Variances of the IAEs with the target temperature for the SHA thermopiles of (**a**) the best case in this study and (**b**) the best case in [16], and (**c**) the thermopile without SHA.

## 4. Conclusions

In this study, the enhancement of IAEs for a CMOS compatible thermopile with special subwavelength hole arrays in the active area was numerically investigated by the finite-difference time-domain method. It was found that the subwavelength rectangular-hole arrays with rectangular-columnar or ellipse-columnar structures in the hole array could be enhanced the absorption efficiency of this thermopile. It was obtained that, for the types of three RCs and three ECs, the enhancements of the IAEs are 14.4% and 15.2%, respectively. Such special subwavelength hole arrays can be improved by the IAE of the CMOS compatible thermopile.

**Author Contributions:** All the authors participated in the design of experiments, analysis of data and results, and the writing of the paper. All authors have read and agreed to the published version of the manuscript.

**Funding:** This research was partially supported by the Ministry of Science and Technology of the Republic of China under Contract No. MOST 108-2221-E-008-042 and MOST 108-2221-E-018-006.

**Acknowledgments:** The authors would like to thank the research which was partially supported by the Ministry of Science and Technology of the Republic of China.

**Conflicts of Interest:** The authors declare no conflicts of interest.

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
