# Peer review of "Infrared Absorption Efficiency Enhancement of the CMOS Compatible Thermopile by the Special Subwavelength Hole Arrays"

_applsci, doi:10.3390/app10082966_

Round 1
Reviewer 1 Report
This paper is a study on improving thermopile performance via subwavelength hole arrays. Optimal arrays were investigated numerically using the finite difference time domain method.
I thought the paper was well organized and the authors work was sound. I would recommend publishing the work in its current form. My only suggestion is that Fig 6 and Table 1 contain the same information so one of them may be eliminated if space is an issue. I think just publishing the table should be sufficient--but just a suggestion.
Author Response
Dear Ms. Chenny Zhao:
Manuscript ID: applsci-777016
Title: Infrared Absorption Efficiency Enhancement of the CMOS Compatible Thermopile by the Special Subwavelength Hole Arrays
Authors: YUN YING YEH, Chih-Hsiung Shen, Chi-Feng Chen *
Thank you for your letter dated April 17, 2020.
Thank you for Referees' comments. We have revised our paper according to the referee’s comments.
- My only suggestion is that Fig 6 and Table 1 contain the same information so one of them may be eliminated if space is an issue. I think just publishing the table should be sufficient--but just a suggestion.
Answer: We have eliminated Fig 6 and the related statements.
- Please check some minor spelling issues ( "the" can be removed in some cases).
Answer: We have modified the manuscript for some minor spelling issues.
- I could not find any measurement results so I suppose there are just simulations. It is hard to verify the paper without measurement results. The technical content is now not very high. Verification would enormously increase the quality of the paper. Seems like an incremental paper of the authors on the subject.
Answer: Thank the unknown Referee for valuable comments. Since we have revealed the verification in the previous work in Ref. 16 and the simulation results are consistent with the measurement results successfully. This research is the extended work based on the results of our previous paper. Therefore, we believe that the simulation results of this study should be reliable. We hope to reveal this interesting phenomenon as soon as possible, so here the simulation results are provided. We agree that verification will greatly improve the quality of paper, but it is difficult to apply for a trial production quota of TSMC 0.35 um CMOS in the near future to manufacture thermopiles.
Sincerely yours
Chi-Feng Chen
Department of Mechanical Engineering/Institute of Opto-Mechatronics Engineering,
National Central University,
No.300, Jhongda Rd., Jhongli City, Taoyuan County 320, Taiwan(R.O.C.)
Phone: +886-3-4267308
Fax: +886-3-4254501
E-mail: [email protected]

Reviewer 2 Report
The paper is well written.
Please check some minor spelling issues ( "the" can be removed in some cases.
I could not find any measurement results so I suppose there are just simulations.
It is hard to verify the paper without measurement results
The technical content is now not very high.
Verification would enormously increase the quality of the paper.
Seems like an incremental paper of the authors on the subject.
Author Response

(The authors gave the same response as above.)
